# Enhanced Light Extraction Efficiency by Self-Masking Technology with Carbonized Photoresist for Light-Emitting Diodes

**DOI:** 10.3390/mi14030534

**Published:** 2023-02-24

**Authors:** Xiu Zhang, Shuqi Li, Baoxing Wang, Baojin Chen, Haojie Guo, Rui Yue, Yong Cai

**Affiliations:** 1School of Nano-Tech and Nano-Bionics, University of Science and Technology of China, Hefei 230026, China; 2Key Laboratory of Nanodevices and Applications, Suzhou Institute of Nano-Tech and Nano-Bionics, CAS, Suzhou 215123, China; 3Ningbo Sky Torch Optoelectronics Technology Co., Ltd., Ningbo 315301, China

**Keywords:** light-emitting diodes, self-masking technology, light extraction efficiency

## Abstract

This work investigates a self-masking technology for roughening the surface of light-emitting diodes (LEDs). The carbonized photoresist with a naturally nano/micro-textured rough surface was used as a mask layer. After growing the Si_3_N_4_ passivation layer on LEDs, the texture pattern of the mask layer was transferred to the surface of the passivation layer via reactive ion beam (RIE) dry etching, resulting in LEDs with nano-textured surfaces. This nano-textured surface achieved by self-masking technology can alleviate the total internal reflection at the top interface and enhance light scattering, thereby improving the light extraction efficiency. As a result, the wall-plug efficiency (WPE) and external quantum efficiency (EQE) of rough-surface LEDs reached 53.9% and 58.8% at 60 mA, respectively, which were improved by 10.3% and 10.5% compared to that of the flat-surface Si_3_N_4_-passivated LED. Additionally, at the same peak, both LEDs emit a wavelength of 451 nm at 350 mA. There is also almost no difference between the I–V characteristics of LEDs before and after roughening. The proposed self-masking surface roughening technology provides a strategy for LEE enhancement that is both cost-effective and compatible with conventional fabrication processes.

## 1. Introduction

GaN-based light-emitting diodes (LEDs) are widely used in various applications, including displays, traffic signals, auto lighting, and backlighting, as a result of the rapid development of LED technology [1,2,3]. Commercial GaN-based blue LEDs with internal quantum efficiencies (IQE) exceeding 80% are now available, but the extraction efficiency of LEDs is still low due to Fresnel reflection and total internal reflection (TIR) at the GaN interface [3]. GaN and air have refractive indices of 2.5 and 1.0, respectively. The critical angle through which light generated in the InGaN/GaN active region can escape is approximately θ_C_ = sin^−1^(n_air_/n_GaN_) = 23° for bare chips [4,5]. That severely limits the LEE of LEDs.

To improve the LEE of LEDs, multiple mainstream strategies, such as high refractive index packaging materials, patterned sapphire substrates (PSS) [6,7], distributed Bragg reflectors (DBR) [8], and Omni-directional reflectors (ODR) [9], have been employed, resulting in a light extraction efficiency (LEE) of more than 50% for LEDs today. Surface roughening [10] was then considered a feasible method for further enhancing the LEE in combination with the above technologies.

Many investigations on surface roughening have been reported [11,12,13,14,15,16,17,18,19]. To enhance the optical and electrical properties of GaN-based LEDs, Alias et al. [15] used NH_4_OH and H_2_O_2_ to roughen N-face GaN substrates, resulting in a uniform distribution of hexagonal pyramidal structures. Na et al. [17] selectively wet-etched the p-GaN layer with hexagonal etch pits using KOH solution, leading to a 29.4% increase in LED light output power. Except for wet etching, dry etching is also capable of generating rough surfaces. Yang et al. [5] used ICP dry etching to roughen the p-GaN layer of LEDs by varying the Cl_2_ concentration in the Cl_2_/BCl_3_/N_2_ gas mixture and adjusting the RF bias power parameters of the inductively coupled plasma (ICP) system. Wang et al. [18] fabricated LEDs with periodic micro-hexagon-patterned ITO using standard photolithography and ICP technology. However, plasma damage occurs in the p-GaN and ITO layers during the dry etching process, worsening the electrical performance of the LEDs [13]. Roughing the LED surface passivation layer appears to be a safer option than dry etching the p-GaN or ITO layer.

In this work, we proposed a LEE enhancement strategy to rough the surface of the Si_3_N_4_ passivation layer by dry etching using a self-masking technology with carbonized photoresist. The self-masking technology involves the development of mask patterns naturally, followed by their transfer to the target layer. When the ion bombardment energy is too high or too long, the photoresist (PR) is prone to wrinkling and carbonization. Previously, wrinkling and carbonization were considered PR defects that should be avoided in general chip processes. However, every question has two sides. The heavily carbonized PR always has a rough nano/micro-mountain texture. As a result, we attempted for the first time to use this phenomenon to achieve the nano-scale roughened surface of LEDs without nano-scale photolithography.

Unlike conventional GaN-based LEDs, Si_3_N_4_ was adopted as the passivation layer rather than SiO_2_. The critical factors for choosing Si_3_N_4_ were its high transmittance (>95%) of light at 450 nm wavelength and refractive index (n = 1.9), which is closer to that of ITO than SiO_2_ (n = 1.48). The natural nano/micro-texture of the mask pattern formed by bombarding the PR with high-energy ions was transferred to the Si_3_N_4_ passivation layer surface by dry etching. The proposed self-masking surface roughening technology used on the passivation layer is noteworthy for its straightforward process, low damage to the device, low cost, and compatibility with conventional manufacturing processes.

## 2. Fabrication

GaN-based LED epitaxial wafers were grown on patterned sapphire substrates by the metal–organic chemical vapor deposition (MOCVD) method. A 3-μm-thick-undoped GaN buffer layer was first grown to relieve mismatch stress, followed by a 2.5-um-thick n-GaN layer and a periodic InGaN/GaN multi-quantum well (MQWs) layer. A 150-nm-thick p-AlGaN layer continued to be deposited as an electron-blocking layer (EBL), followed by a 200-nm-thick p-GaN layer.

The process flow for self-mask-roughened LED devices is shown in Figure 1. A 110-nm-thick ITO layer was grown on the LED epitaxial wafer to enhance current spreading. A 1.2-μm-thick Si_3_N_4_ layer was grown on ITO by the plasma-enhanced chemical vapor deposition (PECVD) method. A 1-μm-thick PR was spin-coated on the Si_3_N_4_ layer as a mask and then bombarded by ion beam etching (IBE) to form nano/micro-mountain-like patterns after carbonization. Next, the carbonized PR was cleaned with RIE dry etching, and the nano-texture pattern was transferred to the surface of the Si_3_N_4_ layer by further etching. The subsequent steps were almost the same as those for conventional LEDs [20]. The epitaxial wafer was selectively etched to the n-GaN layer by the ICP system, followed by the deposition of p-type and n-type metal electrodes.

IBE for PR carbonization and RIE for transferring natural mountain-like texture patterns to the Si_3_N_4_ layer are two important steps in self-masking surface roughening technology. The essential parameters of IBE are shown in Table 1. Notably, IBE technology introduces neutralizing currents to avoid damage to the MQWs layer from the electrostatic field caused by the charge accumulation on the LED surface when the high-energy cations bombard the PR.

The essential parameters of RIE are shown in Table 2. RIE dry etches Si_3_N_4_ mainly with fluorine-based gas, and fluorine active radicals react with Si_3_N_4_ to generate volatile SiF_4_ and N_2_. Further, O_2_ reacts with the polymer formed by the carbonized PR and accelerates the etching speed of Si_3_N_4_ with enhanced isotropic etching. After RIE etching, it can generate a rough surface with a nano-mountain-like texture on the Si_3_N_4_ passivation layer.

For comparison, Si_3_N_4_-passivated LEDs with a flat surface and with a rough surface were prepared to verify the roughening effectiveness of the self-masking technology with carbonized PR, as shown in Figure 2. The size and the wavelength of both LED devices were 1.1 × 1.1 mm^2^ and 450 nm, respectively. A finger-type structure [21] was used for better current spreading, and the back side (sapphire side) of LEDs was deposited with metal as the reflector to enhance the front-side light emission.

## 3. Results and Discussion

Field-emission scanning electron microscope (Nova Nano SEM 450 made by FEI, USA) was used to perform the surface morphologies of carbonized PR, as shown in Figure 3a–c. It can be seen that the surface of the carbonized PR is distributed with a nano/micro-mountain-like texture, which is transferred to the Si_3_N_4_ passivation layer by RIE dry etching as a self-mask. Figure 3d–f depicts the nano-texturized surface morphology of the Si_3_N_4_ layer. The distribution of this texture is random, with a distance of about 200–900 nm between two adjacent mountains. Different from the surface of carbonized PR, a smaller and steeper nano-mountain-like surface was present on the Si_3_N_4_ layer. Figure 3g shows the cross-sectional morphologies of texturized Si_3_N_4_ layer obtained by focused ion beams (FIB) technology. The height of the texture is also random, and the size is about 200–700 nm. Compared to the flat surface, this nano-mountain-like surface is more favorable for light extraction due to its ability to reduce Fresnel emission and top interface TIR through scattering effects [10].

To clarify the effect of self-masking surface roughening technology on LED performance, the light output power as a function of injection current (LOP-I characteristic) for Si_3_N_4_-passivated LEDs with a flat surface and with a rough surface is shown in Figure 4a. Compared to the LED with a flat surface, the LOP of the LED with a rough surface increased by 10.1% at an injection current of 60 mA and by 4.7% at an injection current of 350 mA, respectively. The increase in LOP is mainly attributed to the enhanced LEE. Photons suffering from TIR and planar reflection can be extracted from the nano-textured surface, leading to an increase in light scattering. The results of LOP agree with the observations of SEM.

The injection currents as a function of voltage (I–V characteristic) are shown in Figure 4b. It can be observed that the I–V curves of Si_3_N_4_-passivated LEDs with a flat surface and with a rough surface are almost identical. Both of them exhibited normal p–n diode behavior with a turn-on voltage of about 2.57 V. The inset of Figure 4b shows the current as a function of reverse bias voltage. At −3 V, the leakage currents are 200 nA and 260 nA for Si_3_N_4_-passivated LEDs with a flat surface and a rough surface, respectively. These results indicate that the dry etching used in self-masking surface-roughening technology did not negatively affect the performance of the LEDs.

Figure 5a shows the dependence of the wall-plug efficiency (WPE) of Si_3_N_4_-passivated LEDs with a flat surface and with a rough surface on the injection currents ranging from 0 to 350 mA. WPE was calculated by
(1)$$WPE=\frac{LOP}{IV}$$
where *I* is injection current, and *V* is voltage. The WPE of LEDs with a rough surface reached 53.9% and 44.6% at 60 mA and 350 mA, respectively. In comparison with the LED with a flat surface, the WPE of LED with rough surface increases by 10.3% at an injection current of 60 mA and by 4.5% at an injection current of 350 mA. The WPE enhancement of the surface-roughened LED at a small current (60 mA) is slightly higher than that at a high current (350 mA), which is possibly caused by the self-masking roughening process that slightly relieves the device’s current leakage, as shown in the inset of Figure 4b.

Figure 5b demonstrates the external quantum efficiency (EQE) of Si_3_N_4_-passivated LEDs with a flat surface and with rough surface. EQE was calculated by
(2)$$EQE=\left.\frac{LOP⁄((h\nu ))}{I⁄q}\right.$$
where *h* is the Planck constant, *ν* is the average frequency of emitted light, and *q* is the elementary charge. It can be seen that the EQE falls with increasing injection current, which is a result of the common phenomenon of efficiency droop that exists among nearly all III-nitride-based LEDs [21]. The EQE of LEDs with a rough surface achieved 58.8% of its peak value at 60 mA and 54.2% at 350 mA and exhibited an apparent enhancement compared to the LED with a flat surface, rising to 10.5% at 60 mA and 4.9% at 350 mA, respectively. This EQE enhancement of the surface-roughened LED is comparable to that of WPE and LOP, owing to the fact that the I–V characteristics of LEDs before and after roughening barely change. Further, EQE is the product of IQE and LEE. The IQE depends on the crystal quality and structure of epitaxial wafers [22]. In this work, the same epitaxial wafers were used for all LEDs, so the IQE is the same. The EQE enhancement is mainly attributed to the improvement of LEE due to surface roughening with the self-masking technology.

Figure 6a presents the electroluminescence (EL) spectra of Si_3_N_4_-passivated LEDs with a flat surface and with a rough surface at 350 mA. Both LEDs emit at the same blue peak wavelength of 451 nm because the roughening process does not affect the emission wavelength. Moreover, the emission intensity of the devices with a rough surface is higher than that of the devices with a flat surface, which provides further verification of the enhancement of LEE by rough surfaces.

The far-field emission intensity was measured by the light intensity test system. Figure 6b shows the far-field patterns of Si_3_N_4_-passivated LEDs with a flat surface and with a rough surface. Both LEDs exhibit a Lambeth radiation pattern, and their spatial distribution patterns are almost similar at 350 mA. It indicated that surface roughening with the self-masking technology doesn’t worsen the spatial distribution of light emitted from LEDs.

## 4. Conclusions

Overall, we realized the surface roughness for Si_3_N_4_-passivated LED devices with a peak wavelength of 451 nm by applying self-masking technology. The carbonized PR with nano/micro-mountain-like texture was achieved as a mask by the IBE method, and the LED with a nano-scale rough surface was successfully prepared by RIE dry etching without photolithography. Results showed that the optoelectronic performance of the LED with a rough surface was significantly improved with an almost unchanged I–V characteristic. In contrast to the Si_3_N_4_-passivated LED with a flat surface, the LOP and WPE of LEDs with a rough surface increased by 10.1% and 10.3% at an injection current of 60 mA, respectively. In addition, the maximum EQE of the surface-roughened LED reached 58.8%. It is proven that the rough surface obtained by self-masking technology facilitates the LEE enhancement of LED devices due to the reduction of Fresnel emission and top interface TIR. This work provides a reference method for enhancing the LEE of all III-nitride-based LEDs that is compatible with conventional fabrication processes and low in cost.

## Figures and Tables

**Figure 1 micromachines-14-00534-f001:**
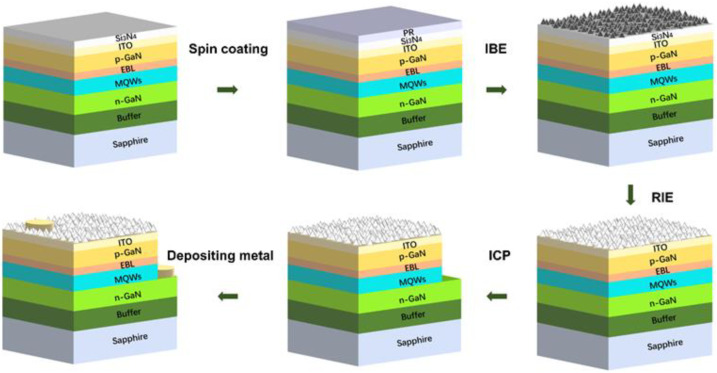
The fabrication process of self-mask-roughened LEDs.

**Figure 2 micromachines-14-00534-f002:**
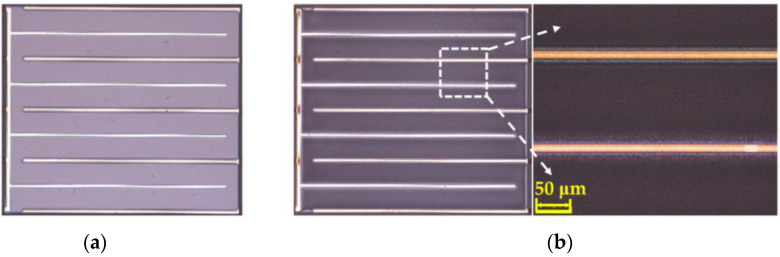
Si_3_N_4_-passivated LEDs with (**a**) a flat surface and with (**b**) a rough surface.

**Figure 3 micromachines-14-00534-f003:**
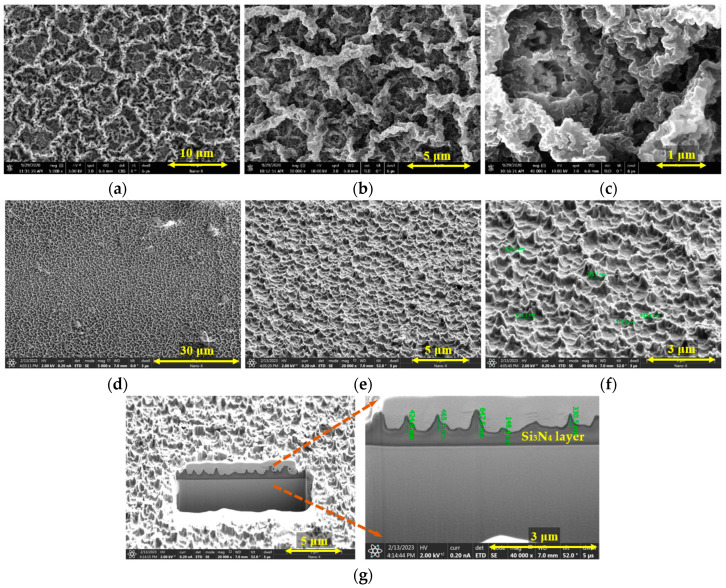
Surface morphologies of carbonized PR: (**a**) 5000×; (**b**)10,000×; (**c**) 40,000×. Surface morphologies of texturized Si_3_N_4_ layer: (**d**) 5000×; (**e**)20,000×; (**f**) 40,000×. Cross-sectional morphologies of texturized Si_3_N_4_ layer: (**g**) 40,000×.

**Figure 4 micromachines-14-00534-f004:**
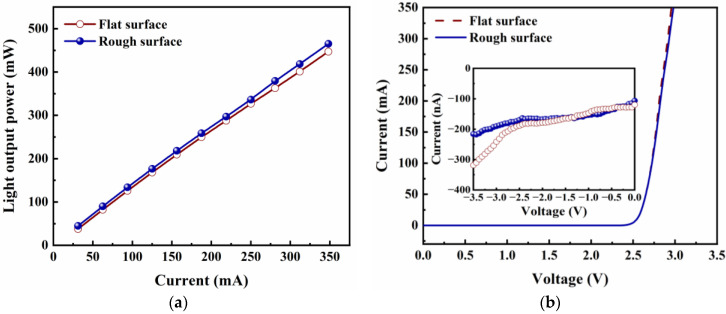
(**a**) LOP−I and (**b**) I−V characteristics of Si_3_N_4_-passivated LEDs with flat surface and with rough surface. The inset shows the dependence of the leakage current on reverse bias voltage of the LEDs.

**Figure 5 micromachines-14-00534-f005:**
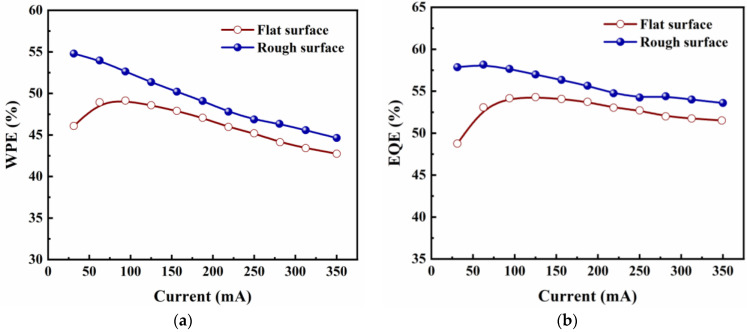
(**a**) WPE-I and (**b**) EQE-I curves of Si_3_N_4_-passivated LEDs with a flat surface and with a rough surface.

**Figure 6 micromachines-14-00534-f006:**
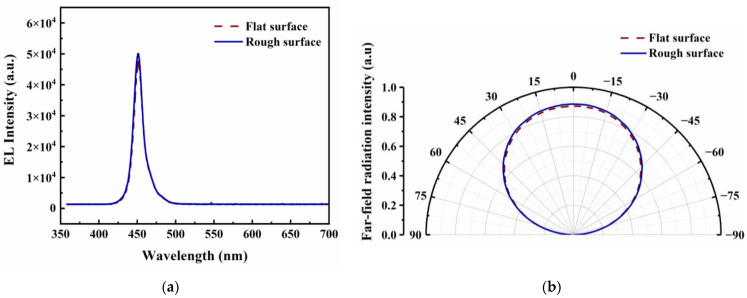
(**a**) EL spectra and (**b**) measured far-field radiation intensity of Si_3_N_4_-passivated LEDs with a flat surface and with a rough surface at 350 mA.

**Table 1 micromachines-14-00534-t001:** Essential parameters of IBE for PR carbonization.

Ion Beam Energy/eV	Ion Beam Flow/mA	Neutralizing Current/mA	Time/min
350	100	120	10

**Table 2 micromachines-14-00534-t002:** Essential parameters of RIE for transferring nano/micro-texture pattern to Si_3_N_4_ layer.

RF/W	CHF_3_/sccm	SF_6_/sccm	Ar/sccm	O_2_/sccm	Time/min
100	13	10	10	10	20

## Data Availability

The data that support the findings are available from the corresponding author upon reasonable request.

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
