# Peer review of "Enhanced Light Extraction Efficiency by Self-Masking Technology with Carbonized Photoresist for Light-Emitting Diodes"

_micromachines, 2023, doi:10.3390/mi14030534_

Round 1

Reviewer 1 Report

In this contribution, authors report the ion beam bombardment induced self-formation of nanostructures in photoresists and its use to further produce nanostructures/roughening of GaN LED surfaces for light extraction enhancement. Although there were quite many other nano/micro structures reported for enhancing GaN LED light extraction, the method reported here is particularly simple (and effective) here. The fabrication and device results are clear and convincing. The article is concise and clear. The work can be recommended for publication after minor revision.

1.  There appear typos in caption of Fig 3. Some (a) (b) (c) should be corrected as (d) (e) (f), I suppose. 

Author Response

Response:

We appreciate the reviewer's attention to details and kind reminder.

    1. The typo in Fig. 3's caption has been corrected. We also double-checked all of the manuscript again for possible typos.

Reviewer 2 Report

1)     In the first paragraph of INTRODUCTION, the manuscript used a theoretical equation to estimate the critical angle. However, this estimation is based on multiple assumptions. In fact unless the conventional LED is used with bare chips, they should be packaged with packaging materials such as silicone. Therefore, the refractive index of the silicone or other materials should be considered even for the technology ten years early. This estimation is far outdated.

2)     In fact, apart from the strategies mentioned in the second paragraph of the INTRODUCTION, there are other well-known strategies for light extraction efficiency improvement. However, there is few notes about this part. It is strongly recommended that the INTRODUCTION should be rewritten.

3)     Although the manuscript claim that the structures is random, however, the manuscript should clear out what can be obtained out from this randomness. In other words, the manuscript should tell the readers the scientific problem or why these structures can help to extract those light rays. Maybe the experiments are time-consuming, numerical simulations are nice tools to help uncover what is hidden behind. Simply attributing the reasons to TIR breaking is not acceptable since at least twelve years early engineers already widely report this conclusion.

4)     The manuscript should put in more characterization results, for example the light intensity distribution curves (LIDC). In fact, not only the EQE is improved, the LIDC may also be changed as well as other optical properties. This manuscript should present the interesting findings in a more thorough way and not in the present form.

Author Response

Response:

We appreciate the reviewer for his kind recommendations and we regret there were problems with the English. The manuscript has been carefully revised by a native English speaker to improve the grammar and readability.

For comment 1:We agree with the reviewer that the manuscript needs to consider that, in fact, The light extraction efficiency (LEE) of LEDs is far beyond the theoretical value of bare chips due to high refractive index packaging materials. So, an update has been made to the manuscript, and the argument that the LEE of GaN-based LEDs is limited to about 4% has been removed.

For comment 2:The introduction has been rewritten. Mainstream strategies for improving light extraction efficiency have been included, such as highly refractive packaging materials, PSS substrate, and DBR.

For comment 3:It is a very interesting and critical issue for LED surface roughness, which is also very attractive to us. Actually, we have attempted  to build a 3D random structure model for LEE simulation. However, we failed. There are too many random structures that should be considered, and the workstation that we use can’t support the huge computation. We think it should be a long-term research project. On the other hand, this work focuses on how to realize a simple and cost-effective surface roughness process. To our best knowledge, the classical TIR-breaking explanation for LEE enhancement of a surface-roughened LED chip is still valid. It is reasonable to attribute the LEE enhancement to TIR breaking in this work.

For comment 4:Thanks for the reviewer’s suggestion. It is our next task to comprehensively investigate the properties of the LED roughened by self-masking technology to verify the possible industry application. We are currently unable to complete all of these tasks due to a lack of resources, such as LIDC test equipment. This work focuses on LEE improvement technology, and we think the EQE, LOP, and WPE data are adequate to support our conclusion.

Reviewer 3 Report

In this work, authors reported the LEE improvement of blue LED by PR roughness. This work could be accepted, after some minor revision:

1) In the first paragraph of 'Introduction', author said "... which limits the light extraction efficiency (LEE) of GaN-based LEDs to about 4% ". I understand that the LEE of visible LED was limited by the refraction index contrast, however, the value of '4%' should be exaggerated by authors. The EQE of visible LED, especially blue ones, has exceeded 40%, how could the LEE be so low of ‘4%’?

2)  In Fig.5, why the WPE~J curve of 'flat surface LED' had a maximum at 100mA, while 'rough LED' did not have this? Please give a brief discussion. 

Author Response

Response:

We appreciate the reviewer’s comment.

For comment 1: In the first paragraph, "the light extraction efficiency (LEE) of GaN-based LEDs is limited to 4%" was cited from Ref. 4 (Yang, Y.; Ren, Y.; et al., Light output enhancement of GaN-based light-emitting diodes by maskless surface roughening. Microelectronic Engineering, 2015, 139, 39-42.), where 4% is the theoretical calculated value when the light is directly incident into the air from the active region of GaN-based LEDs. We agree with the reviewer that the light extraction efficiency of blue LEDs is now far beyond 4% because many methods, such as PSS substrates and high refractive index packaging materials, contribute significantly to the LEE enhancement.

For comment 2: With respect to "flat LED the WPE~J curve has a maximum value at 100 mA, but rough LED does not", we think that the roughing process caused the left shift of the WPE peak of rough LED. Fig. 4(b) has confirmed that the device's current leakage is slightly alleviated due to the self-masking roughening process. The mechanism shall be further investigated.

Reviewer 4 Report

Dear autor, 

compact work on improving LED performance by surface roughening technology. I hope that in the future we will also read about the reliability of the new promising layer structure. 

Best regards, 

the reviewer 

Author Response

Thanks for your approval. We will work harder to do a good job.

Reviewer 5 Report

Figure 3 + text: " the height 117 texture is about 200~300nm" - how this can be seen? The image of the chip (break) must be provided. Please also show "the distance ... about 118 200~800nm" - show this distance over the Figure    Fig 4a: I can see the difference between two curves, but not so obvious; to be sure that the effect occurs indeed, please add the error bars (and reproduce the experiment).   "EQE was defined as (eq (2))" - was it directly measured using the integration sphere?   Please add the EL spectra of the LEDs - both with and w/o the surface roughning.   Now the main question is. As far as I understand, the difference (clearly visible) in the Fig 5 is obtained by the division of the curves from Figs 4 a and b. Furst, in the Fig 4b, we can only see the data corresponding to the current of below 60 mA, while in Fig 5 it is up to 350 mA. Please show the full IV data used to build the WPE plot. Second, in Figs 4 a and b, both blue and red curves (flat and rough surface) almost coincide - in fig 4a, the difference can be seen, though. small one, but in Fig 4b the curves cannot be even distinguished. From where such a difference in Fig 5 originate? Please ensure that it is not a result of a "zero-by-zero division". And add error bars.

Author Response

Response:We appreciate the reviewer for raising these questions.

  1. In the revised manuscript, we add the cross-sectional view of texturized passivation layer obtained by focused ion beams (FIB) technology in Fig 3g.

Fig 3g cross-sectional view of the LED sample with texturized surface

  1. The insignificant difference of the LOP-I curves in Fig 4 while the obvious difference of the WPE-I curves and EQE-I curves in Fig 5 are caused by the different range of values set on the Y-axis. In addition, the LOP rises extremely fast as the current increases at 0~350mA. The improvement of LOP by the roughening technology does not appear to be significant, due to the fact that this work was done on a high base of LOP. We accomplished what was difficult to do even if the improvement was not large.
  2. EQE is calculated by equation 2, and the value of LOP in equation 2 is measured by the integrating sphere.
  3. We show the full I-V data at 0-350mA in the revised manuscript of Fig 4(b).

Fig 4b I-V characteristics of Si3N4 passivated LEDs with flat surface and with rough surface

  1. We add the EL spectra of LEDs w/o surface roughness in Fig 6a to further verify the enhancement of LEE by rough surfaces.

Fig 6a EL spectra of Si3N4 passivated LEDs with flat surface and with rough surface

  1. The I-V, LOP-I and WPE-I curves with error bars are shown below. The data in the manuscript are the typical best values of 11 samples (including 5 reference samples).

Round 2

Reviewer 2 Report

It is nice to find that the authors greatly changed the INTRODUCTION, and this part is fine with me. 

Unfortunately, apart from the INTRODUCTION, I am supersized to find that all technical contents related to both the simulation or characterization of the LEDs were completely neglected by the manuscript. In fact, widely used commercial softwares such as Tracepro or even Lumerical FDTD can accomplish the simulation work nicely. Simply replying in the REPLY TO THE COMMENTS that “Actually, we have attempted to build a 3D random structure model for LEE simulation. However, we failed. There are too many random structures that should be considered, and the workstation that we use can’t support the huge computation. We think it should be a long-term research project.” will be very hard to persuade the readers. It is also interesting to find in the REPLY TO THE COMMENTS that “We are currently unable to complete all of these tasks due to a lack of resources, such as LIDC test equipment.” In fact, what you need is just an integrated sphere. 

It is regretted to say that by refusing to revise the technical content, the creativity of using random structures to improve the LEE of the conventional LED will be greatly jeopardized. 

Author Response

Response:We appreciate the reviewer for further feedback.

  1. For 3D simulation:

The surface and cross-sectional view of the LED sample with texturized surface were performed by FIB-SEM, as shown in       Fig 3(d-g). The size and height of the texture pattern are random, only with an approximate range. That brings a great challenge to the 3D simulation. The results of our constructed model used Lumerical FDTD soft can't effectively match with the experiment. The simulation may need some more specialized help. We expect to do a better job of this in the future.

Fig 3d-g surface and cross-sectional view of the LED sample with texturized surface

  1. For LIDC test:

In the revised manuscript, the far-field emission intensity was measured by the light intensity test system. Figure 6(b) shows the far-field patterns of Si3N4 passivated LEDs w/o the rough surface. Both LEDs exhibit a Lambeth radiation pattern, and their spatial distribution patterns are almost similar at 350 mA. It indicated that surface roughening with the self-masking technology doesn’t worsen the spatial distribution of light emitted from LEDs. In addition, the light output of both LEDs is close to each other at relatively high currents, thus the difference is not significant.

Fig 6b Measured far-field radiation intensity of Si3N4 passivated LEDs with a flat surface and with a rough surface at 350mA

Reviewer 5 Report

Could the authors please also add the error bars on the last Fig.